# Neonatal mortality clustering in the central districts of Ghana

**George Adjei** [1]*, **Eugene K. M. Darteh**[2], **David Teye Doku**[2]

**1** University of Cape Coast, Department of Community Medicine, Cape Coast, Ghana, **2** University of Cape Coast, Department of Population and Health, Cape Coast, Ghana

\* George.adjei2@ucc.edu.gh

## Abstract

### Introduction

Identifying high risk geographical clusters for neonatal mortality is important for guiding policy and targeted interventions. However, limited studies have been conducted in Ghana to identify such clusters.

### Objective

This study aimed to identify high-risk clusters for all-cause and cause-specific neonatal mortality in the Kintampo Districts.

### Materials and methods

Secondary data, comprising of 30,132 singleton neonates between January 2005 and December 2014, from the Kintampo Health and Demographic Surveillance System (KHDSS) database were used. Verbal autopsies were used to determine probable causes of neonatal deaths. Purely spatial analysis was ran to scan for high-risk clusters using Poisson and Bernoulli models for all-cause and cause-specific neonatal mortality in the Kintampo Districts respectively with village as the unit of analysis.

### Results

The study revealed significantly high risk of village-clusters for neonatal deaths due to asphyxia (RR = 1.98, p = 0.012) and prematurity (RR = 5.47, p = 0.025) in the southern part of Kintampo Districts. Clusters (emerging clusters) which have the potential to be significant in future, for all-cause neonatal mortality was also identified in the south-western part of the Kintampo Districts.

### Conclusions

Study findings showed cause-specific neonatal mortality clustering in the southern part of the Kintampo Districts. Emerging cluster was also identified for all-cause neonatal mortality. More attention is needed on prematurity and asphyxia in the identified cause-specific neonatal mortality clusters. The emerging cluster for all-cause neonatal mortality also needs more attention to forestall any formation of significant mortality cluster in the future. Further

**Data Availability Statement:** All relevant data are within the paper and its Supporting Information files.

**Funding:** The authors received no specific funding for this work.

**Competing interests:** The authors have declared that no competing interests exist.

research is also required to understand the high concentration of prematurity and asphyxiated deaths in the identified clusters.

## Introduction

The decline of neonatal mortality rates have been slower than those of under-five mortality globally [1]. It is projected that, 27.8 million neonates will die between 2018 and 2030 if the current trend continues [2]. Globally, neonatal mortality rate fell by 51% between 1990 and 2017 as compared to 63% of postneonatal mortality rate within the same period [2, 3]. This 51% reduction in neonatal mortality translates into 37 deaths per 1000 live births in 1990 to 18 deaths per 1000 live births in 2017 [2]. In 2017 alone, 2.5 million newborns died within the first month of life which was approximately 7,000 deaths per day [3]. Meanwhile, the Sustainable Development Goal (SDG) 3 target of reducing neonatal mortality rate to 12 deaths per 1000 live births will require more than 60 countries to accelerate their progress in order to achieve the target by 2030 [2, 3].

Globally, sub-Saharan Africa has the highest rate of neonatal mortality and accounts for 38% of global neonatal deaths [4, 5]. According to global estimates report, sub-Saharan Africa accounted for 41% of total burden of neonatal deaths in 2017 alone [2]. In Ghana, 48% of under-five mortality occurs in the neonatal period and the rate of neonatal mortality is twice that of postneonatal mortality [6]. Unless concerted efforts are made, Ghana will not be able to achieve SGD 3.

There are many risk factors of neonatal mortality and that includes birth spacing, maternal age at birth, asphyxia, pneumonia, meningitis, low birth weight, sepsis and region in which the child was born [7–11]. Although there are many risk factors of neonatal mortality, each geographical area has specific risk factors that predisposes neonates to death [12, 13]. It is important therefore to consider geographical setting in the study of neonatal mortality to consequently provide the appropriate intervention. Besides, population-based interventions are good at preventing many people from morbidities and mortalities but can be a huge undertaken and expensive. Hence, when population-based interventions for neonatal mortality are expensive to implement, it is necessary to identify high-risk mortality clusters and provide them (clusters) with the needed resources which are less expensive and more useful to poor resource settings in sub-Saharan Africa [14].

In Ghana, spatial clustering of under-five mortality has been investigated in a couple of studies [15, 16]. Nettey et al. [16] and Adjuik et al. [15] studies used villages as units for clustering and were able to identify high-risk areas for under-five mortality clustering. Clustering for neonatal mortality which constitute highest proportion of under-five mortality in Ghana [6] was not considered in those studies. However, MDG 4 of reducing under-five mortality by two-thirds was not achieved by Ghana and Ghana is currently striving to achieve SDG 3 which has a target of reducing neonatal and under-five mortality rates to 12 and 25 deaths per 1000 live births respectively. Hence, knowledge of all-cause and cause-specific neonatal clustering will make intervention for neonatal mortality more focused and less expensive [17]; and subsequently contribute to the achievement of SDG 3 in Ghana.

The Kintampo North Municipality and South District (Hereinafter referred to Kintampo Districts) in the middle belt of Ghana also experiences a slower decline in neonatal mortality and has 42% of neonatal mortality occurring in the districts [18, 19]. Routine data is collected on core demographic data using the health and demographic surveillance system. In addition,

houses are geocoded and therefore make it feasible for neonatal mortality data to be used for spatial studies to expand the literature in this area. This study therefore aims to identify villages with high risk of all-cause and cause-specific neonatal mortality in the Kintampo Districts.

## Materials and methods

### Study area

This study used a secondary dataset from Kintampo Health and Demographic Surveillance System (KHDSS) which covers virtually the whole of Kintampo North Municipality (located between latitudes 8˚45'N and 7˚45'N and Longitudes 1˚20'W and 2˚1'E) and Kintampo South District (located between 1º 20' West and 2º10' East and latitude 8º 15' North and 7º 45' South) [16, 18]. Both the Municipality and District together have a total surface area of 7,162 km². The KHDSS area is also located in the Brong Ahafo Region of Ghana.

The total population and households in the KHDSS area in 2014 were 145,000 and 32,000 respectively. The main indigenous ethnic groups are the Bonos and the Mos. There is however a large immigrant population from the three Northern Regions (Dagaaba, Dagomba and Konkomba) who are generally farmers. Some Dangbe and Ewe immigrants are mainly fisher folks and are settled along the banks of the Black Volta. The settlements are mostly concentrated in the Southern part and along the main trunk road linking the two District capitals (Kintampo and Jema) to Northern Region. Community members engage in farming of predominantly maize, yam and cassava. Livestock rearing of cattle, sheep, goats and birds are also common.

Each district in the KHDSS area has a hospital and other health facilities such as health centres, CHPS Compounds and maternity homes that perform family planning services as well as maternal, child and neonatal services. Outreach health services are also performed by CHPS compounds within the communities [20].

### Determination of causes of deaths

Causes of neonatal deaths from the secondary data were determined by using a validated verbal autopsy (VA) tool with a sensitivity greater than 60% for all major causes [21]. Specificity was found to be 76% for birth asphyxia and greater than 85% for prematurity and infection [21]. The VA tool had a battery of questions together with a narration section where the immediate caregiver of the neonates gave a vivid accounts of what might have led to the death of the neonate [22]. The VA interviews were conducted by well-trained field supervisors of KHRC and it was ensured that the interviews were conducted as closely as possible to the day of death to reduce recall bias. Because field supervisors training includes psychological aspect, they were able to handle emotional and psychological trauma during the interview. Data collection were rigorously supervised and 5% of the interviews were repeated to ensure data quality. In cases where discrepancies were detected, interviews were repeated. Three clinicians trained on coding of verbal autopsies reviewed the same VA questionnaire independently to assign the possible cause of death in accordance with 3-digit code of the International Statistical Classification of Diseases and Health-Related Problems [18, 22]. The possible cause of neonatal death is established if there was concordance in the review diagnosis of at least two clinicians. In addition, VA questionnaires were submitted to two additional clinicians for review when there was discordance among all the first three clinicians. The cause of neonatal death was declared indeterminate where consensus could not be reached with regards to possible cause of death. Moreover, the possible cause of neonatal death was declared unknown where there was little or no information to making it possible to establish the possible cause of death.

## Classification for neonatal causes of death

Neonatal deaths in the secondary dataset were classified into six major categories namely congenital abnormality, prematurity, birth asphyxia, infection (sepsis, meningitis, pneumonia, septicaemia, tetanus, diarrhea and other neonatal infections), other specific causes of death (malaria, injury, infant haemorrhage and other specific causes excluding the first four major causes) and unexplained cause of death. This classification method was based on Neonatal and Intrauterine Death Classification according to Etiology (NICE) and the WHO Child Health Epidemiology Reference Group (CHERG) and has been described elsewhere [23]. Table 1 depicts the classification of the six major categories. Hence the cause-specific neonatal deaths for this study were six.

## Data management

This study used secondary data from the KHDSS database comprising of all singleton neonates (dead or alive) and their corresponding VA dataset between January 2005 and December 2014. The data were assessed for any discrepancies and errors found were corrected with the assistance of a Data Manager from Kintampo Health Research Centre (KHRC). The cleaned dataset was exported to SaTScan software for spatial analysis.

## Spatial analysis

Due to low proportions of neonatal mortality in each year, purely spatial analysis based on the total mortality over the study period was employed. The Poisson and Bernoulli models were used to run a purely spatial analysis scanning for clusters (village or group of villages) with high rates of all-cause and cause-specific neonatal mortality respectively. Regarding the Bernoulli model, definition of cases were all neonatal deaths that were caused by a specific morbidity. Controls were all other deaths caused by other morbidities together with other neonates who were alive. Scanning was set to identify villages with high proportion of all-cause and cause-specific neonatal deaths because the interest was to search for a village or a group of

**Table 1. Classification of 6 major causes of death.**

| Number | Neonatal death | Infant born alive and cried, moved or breathed after birth and then died within the first 28 days of life |
|---|---|---|
| 1 | Congenital abnormality | Neonatal death due to one or more of the following: Major or lethal congenital abnormality Unspecified; specific abnormality e.g neurological, neural tube defect |
| 2 | Prematurity | Neonatal death due to one or more of the following: Severe immaturity (<33 weeks), birthweight<1.8 kg where gestation is unknown, specific severe complications of prematurity such as surfactant deficiency or necrotising enterocolitis |
| 3 | Birth asphyxia | Neonatal death in an infant ≥33 weeks of gestation due to one or more of the following: Obstetric complications, maternal haemorrhage, or clinical diagnosis of birth asphyxia (no cry soon after birth plus either convulsion/spasms or not able to suckle normally after birth) |
| 4 | Infection | Neonatal death in an infant ≥33 weeks of gestation due to one or more of the following: Tetanus, meningitis, pneumonia, diarrheoa, septicaemia, other infection |
| 5 | Other | Neonatal death in an infant ≥33 weeks of gestation due to a cause not included in first 4 selected causes including: Accident/injury, infant haemorrhage respiratory distress syndrome, severe neonatal jaundice |
| 6 | Unexplained | Neonatal death due to unknown cause including sudden infant death syndrome |

**Table 2. Identified potential clusters and their villages in the Kintampo Districts.**

| Cluster number | Village(s) within a cluster |
|---|---|
| 1 | Bawa Akura 1, Krabonso, Aworata, Bobrobo, Akruma, Adiembra, Nante, NanteZongo, Tanokrom, Hyireso, Ampoma |
| 2 | Abom Basare, Abom Kokonba, Jerusalem, Apesika, Anokyekrom, Akora Nkwanta, Akora, Nyamebekyere, Brechakrom, Attakrom, Asuogya No. 1 |
| 3 | Yaara |
| 4 | Techira No. 1 |

villages with proportion of all-cause and cause-specific neonatal mortality that was higher than average. Geographic overlap was not considered in the setting so secondary clusters did not overlap the most significant (or likely) cluster. Maximum cluster size was set to 50% of the total population at risk in order to scan for small to large clusters. In order to ensure sufficient statistical power, the number of Monte Carlo replications was set to 999 and clusters with p<0.05 were considered as statistically significant.

## Ethical consideration

This study which has minimal risk was carried out by analysing secondary dataset. Study participants confidentiality was ensured since unique ID codes were used to identify them. The core KHDSS activities (births, deaths, migration) have been given ethical clearance from the Research Development and Division (of the Ghana Health Service) Ethical Review Board.

## Results

### All-cause neonatal mortality clustering

The Poisson model was used in this analysis that involved 30,132 population at risk with 634 neonatal deaths. Table 2 depicts the villages in each all-cause neonatal mortality cluster while Table 3 presents the results of the spatial scan statistics using the Poisson model. No statistically significant cluster was detected in the model (Table 3). However, four potential clusters were detected for all-cause neonatal mortality with the first cluster (most likely or primary cluster) containing eleven villages, second cluster (secondary cluster) containing eleven villages and the third (secondary cluster) and fourth (secondary cluster) clusters containing only one village (Table 3). The first potential cluster which was in the south-western part of the Kintampo HDSS area had a radius of 8.22 km with 71 all-cause neonatal deaths and a relative risk of 1.51 (p = 0.333). The second potential cluster can also be located in the south-eastern part of the KHDSS area with a radius of 10.83 km, forty-eight cases and a relative risk of 1.56 (p = 0.640). The third potential cluster was in the north-western part of the KHDSS area but since it contains only one village (Yaara) and the villages are represented by their centroids, its radius was denoted as 0 km. The number of all-cause neonatal deaths identified in this cluster

**Table 3. Detected clusters from purely spatial analysis using the Poisson model.**

| Cluster type | No. of villages within a cluster | Radius (km) | Observed cases | Expected cases | Relative risk | p-value |
|---|---|---|---|---|---|---|
| Most likely | 11 | 8.22 | 71 | 48.96 | 1.51 | 0.333 |
| Secondary | 11 | 10.83 | 48 | 31.69 | 1.56 | 0.640 |
| Secondary | 1 | 0.00 | 5 | 1.43 | 3.51 | 0.967 |
| Secondary | 1 | 0.00 | 5 | 1.56 | 3.23 | 0.993 |

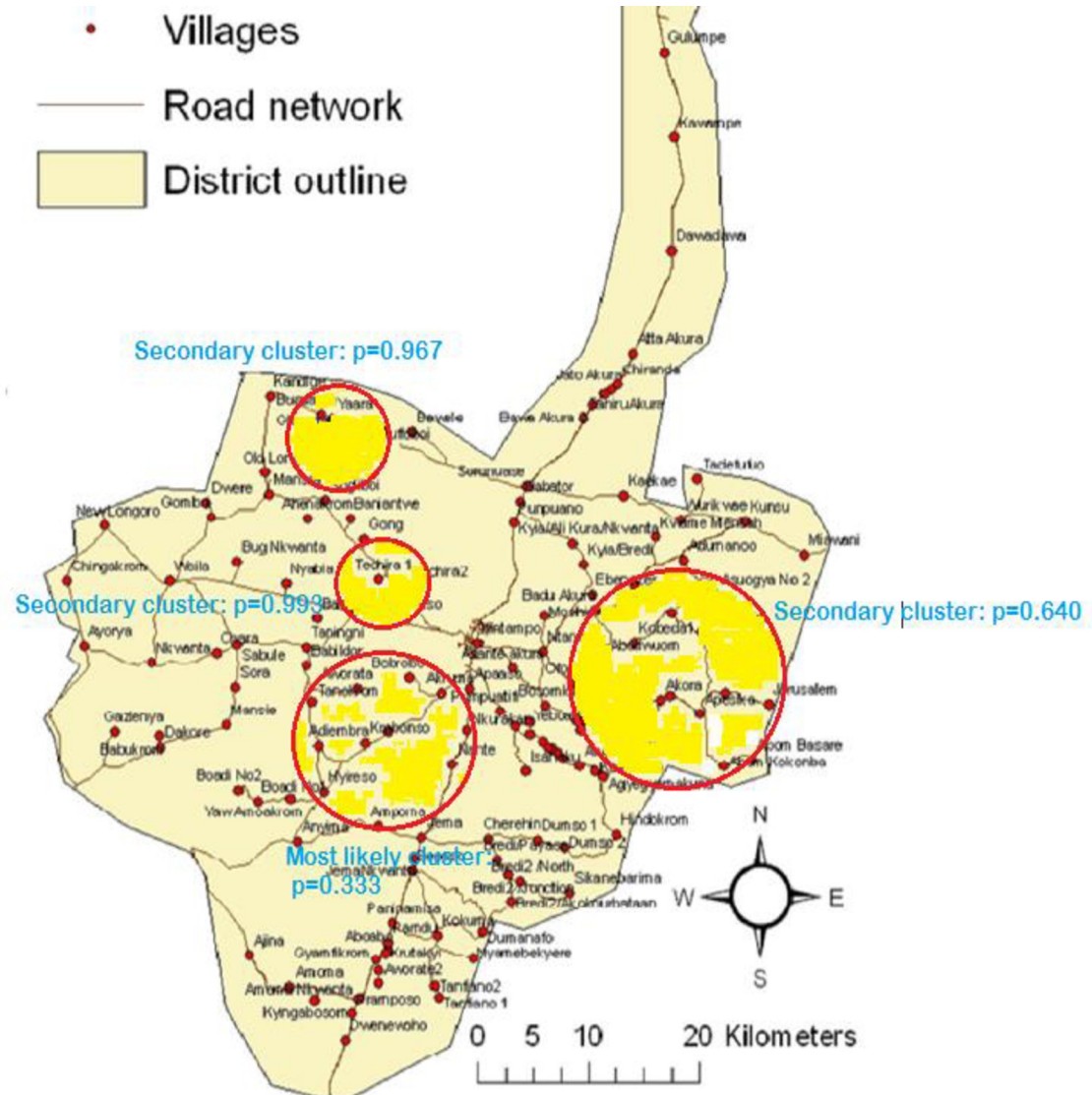

**Fig 1. Most likely and secondary clusters for all-cause neonatal mortality in the Kintampo Districts.**

was five with a relative risk of 3.51 (p = 0.967). The fourth cluster also contains only one village (Techira No.1) and had five cases with a relative risk of 3.23 (p = 0.993). This cluster is located in the north-western part of the KHDSS area. Fig 1 depicts the four potential clusters in the Kintampo Districts.

## Cause-specific neonatal mortality clustering

Table 4 depicts the villages in each detected cluster and Table 5 summarises the cause-specific neonatal mortality results from the Bernoulli model. The analysis involved 474 neonatal deaths attributed to specific causes. This constitutes 74.8% of the 634 neonatal deaths considered in the study. The missing data were due to verbal autopsy interviews which could not be conducted as a result of out-migration and also outstanding verbal autopsy coding yet to be done by physicians.

**Table 4. Detected clusters and villages for neonatal cause-specific mortality.**

| Cause-specific mortality | Cluster number | Village(s) within a cluster |
|---|---|---|
| Asphyxia | 1 | Anyima, Alhassan Akura No. 2, Yaw Amoakrom, Boadi No1, Hyireso, Boadi No. 2, Ampoma, Adiembra, Krabonso, Beposo, Jema Nkwanta, Ajina, Paninamisa, Jema, Mansie, Pamdu, Tanokrom |
| | 2 | Asuogya No 1, Attakrom, Brechakrom |
| Congenital | - | No cluster detected |
| Infection | 1 | Nante Zongo, Nante, Pumpuatifi |
| | 2 | Kandige, Busuama, Yaara, Old Longoro, Mansra, Gbuonyonga, Tuffoboi, Sogliboi, Ntareban, Dwere, Ahenakrom, Gomboi, Baniantwe, Bewele, Bug Nkwanta, Gong, Nyabia, Weila, New Longoro, Techira No.1, Basabasa, Asantekwa, Techira2, Sabule, Yabraso, Taningni, Chara, Babiledor Konkonba, Babildor, Soronuase, Babator, Chingakrom, Punpuano, Nkwanta, Sora, Aworata, Tanokrom, Ayorya, Bobrobo, Tahiru Akura |
| | 3 | Asuogya No. 1 |
| Prematurity | 1 | Bawa Akura 1, Krabonso, Aworata, Bobrobo, Akruma, Adiembra, Nante, Nante Zongo |
| | 2 | Kwabia, Yepemso, Oforikrom, Drepo, Yeboah, Abudwuom, Fokuokrom, Nyamebekyere, Brechakrom, Attakrom, Akora, Bosomkai, Kobeda1, Asuogya No. 1, Akora Nkwanta, Ntankro |
| | 3 | Ajina, Amoma |
| Other | 1 | Brechakrom, Attakrom, Nyamebekyere, Asuogya No 1, Fokuokrom, Akora, Akora Nkwanta, Agyegyemakunu, Kofiekrom, Kwabia |
| | 2 | Sora, Mansie, Chara, Sabule, Babiledor Konkonba, Babildor, Tanokrom, Taningni, Nkwanta, Babukrom, Dakore, Boadi No. 2, Adiembra, Basabasa, Boadi No. 1, Nyabia, Yaw Amoakrom, Aworata, Weila, Bug Nkwanta, Gazienya, Hyireso, Asantekwa, Krabonso, Ayorya, Alhassan Akura No. 2, Bawa Akura 1, Anyima, Yabraso, Gomboi, Bobrobo, Techira No. 1, Ahenakrom, Dwere |
| | 3 | Atta Akura, Chiranda |
| Unexplained | 1 | Gong, Baniantwe, Techira No. 1 |
| | 2 | Agyegyemakunu, Kofiekrom |

**Table 5. Detected clusters for neonatal cause-specific mortality using the Bernoulli model.**

| Cause-specific mortality | Cluster type | No. of villages within a cluster | Radius (km) | Observed cases | Expected cases | Relative risk | p-value |
|---|---|---|---|---|---|---|---|
| Birth asphyxia | Most likely | 18 | 12.23 | 56 | 32.62 | 1.98 | 0.012 |
| | Secondary | 3 | 1.23 | 5 | 1.07 | 4.77 | 0.608 |
| Congenital | No cluster detected | - | - | - | - | - | - |
| Infection | Most likely | 3 | 3.46 | 10 | 3.94 | 2.68 | 0.630 |
| | Secondary | 41 | 28.09 | 29 | 18.07 | 1.79 | 0.650 |
| | Secondary | 1 | 0 | 2 | 0.24 | 8.60 | 0.960 |
| Prematurity | Most likely | 8 | 7.03 | 10 | 2.28 | 5.47 | 0.025 |
| | Secondary | 16 | 6.47 | 6 | 2.11 | 3.16 | 0.891 |
| | Secondary | 2 | 4.57 | 4 | 1.43 | 3.00 | 0.993 |
| Other | Most likely | 10 | 4.83 | 7 | 2.37 | 3.20 | 0.810 |
| | Secondary | 34 | 16.35 | 15 | 7.85 | 2.20 | 0.810 |
| | Secondary | 3 | 3.00 | 4 | 0.97 | 4.35 | 0.850 |
| Unexplained | Most likely | 3 | 3.69 | 3 | 0.21 | 15.33 | 0.222 |
| | Secondary | 2 | 0.91 | 3 | 0.41 | 7.82 | 0.689 |

Two clusters were identified for birth asphyxia related deaths. All the clusters were located in the southern part of the KHDSS area with the first cluster consisting of eighteen villages while the second cluster was made up of three villages. The first cluster (most likely cluster) which covered a radius of 12.23 km was statistically significant (p = 0.012) with fifty-six cases and relative risk of 1.98. On the contrary, the second cluster (secondary cluster) was not statistically significant (p = 0.608) but had a relative risk of 4.77 and covered a radius of 1.23 km. Fig 2 depicts the primary and secondary clusters for deaths due to birth asphyxia.

There was no cluster detected for the neonatal deaths attributed to congenital cases as there were just two cases in total. Three clusters were detected for infections but none of them was statistically significant. The first cluster (most likely cluster) which contains three villages and had a radius of 3.46 km was in the south-western part of the KHDSS. Ten observed infection-related deaths were identified in the cluster which had a relative risk of 2.68 (p = 0.63). The second potential

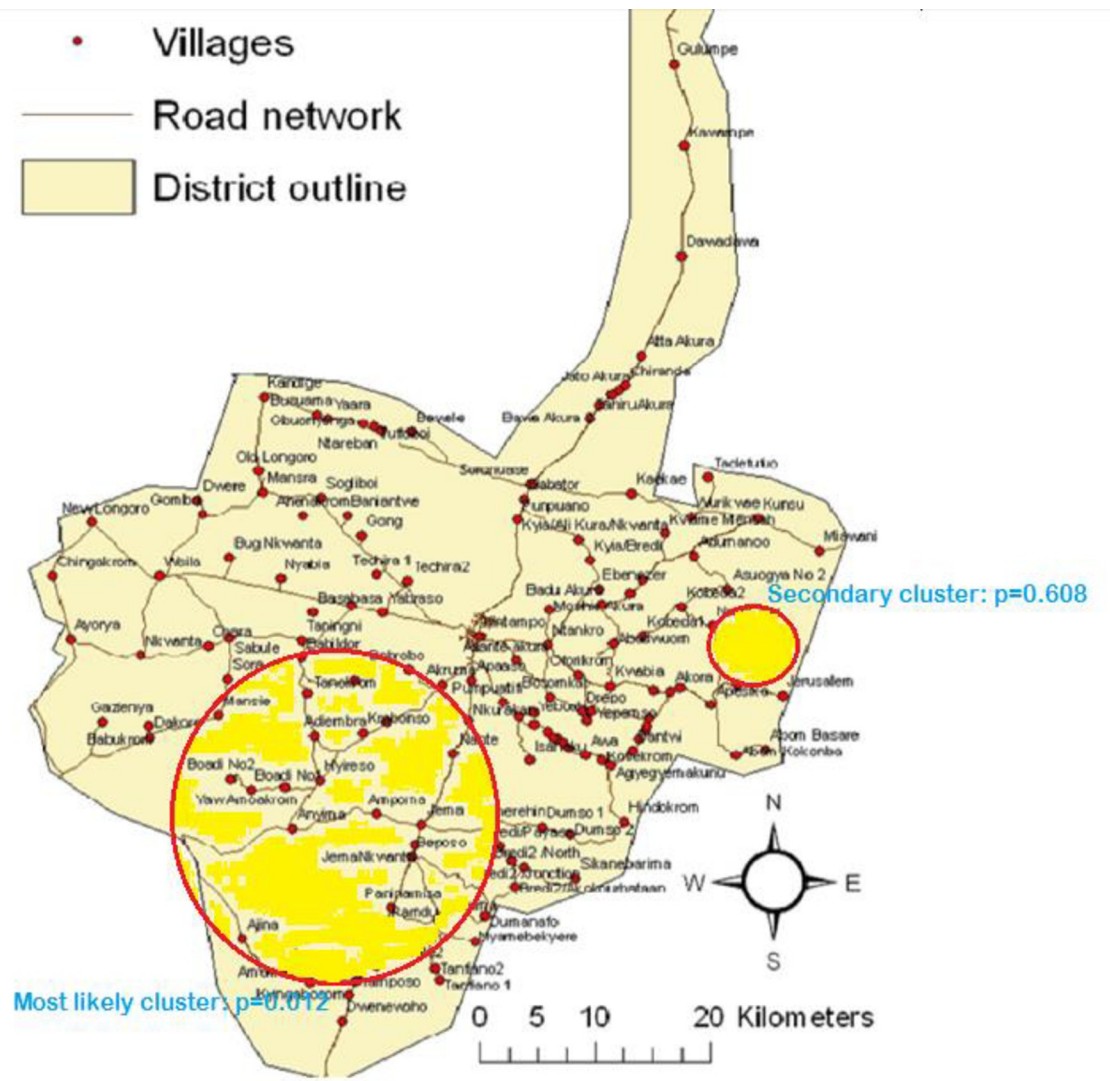

**Fig 2. Most likely cluster and secondary cluster for asphyxiated deaths in the Kintampo Districts.**

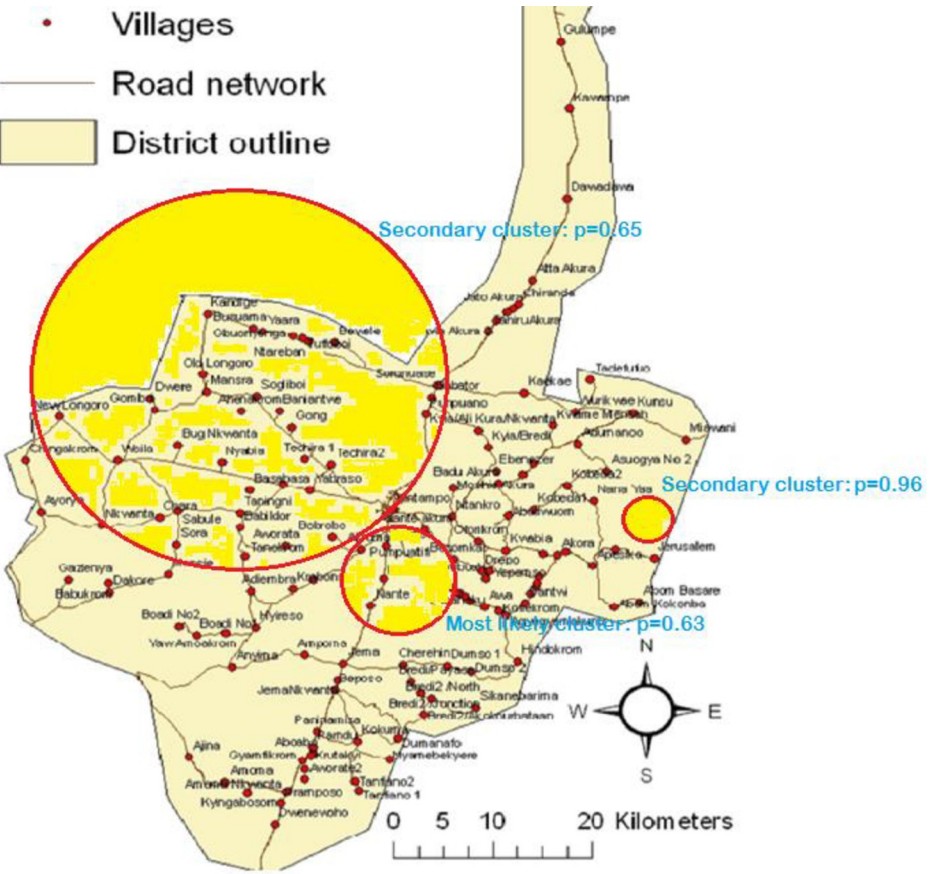

**Fig 3. Most likely and secondary clusters for neonatal deaths due to infections in the Kintampo Districts.**

cluster (secondary cluster) for infection-related deaths contains forty-one villages covering a radius of 28.09 km in the central part of the KHDSS area. The number of cases in this cluster was 29 and had a relative risk of 1.79 (p = 0.65). The third cluster (secondary cluster) for neonatal deaths attributed to infection was also in the north-eastern part of the KHDSS area and also contains only one village (Asuogya No. 1). It had two cases and a relative risk of deaths due to infections to be 8.60 (p = 0.96). Fig 3 shows the potential clusters for neonatal deaths attributed to infections.

Three clusters were detected in the southern part of the KHDSS area as a result of deaths due to prematurity. The first cluster was significant (p = 0.025) and consists of eight villages covering a radius of 7.03 km. Ten deaths were identified from premature births and the relative risk of cases in the cluster was 5.47. The second cluster had sixteen villages, 6.47 km radius, ten cases and a relative risk of 3.16 (p = 0.891). In addition, the third cluster consists of two villages and covering a radius of 4.57 km. Four cases of premature-related deaths were identified in the cluster and a relative risk of 3.00 (p = 0.993) being the risk of cases. The primary and secondary clusters for premature deaths are depicted in Fig 4.

Regarding neonatal deaths attributed to other causes such as infant haemorrhage, injury etc., three potential clusters were identified; the first cluster (most likely cluster) was in the southern part, the second cluster (secondary cluster) in the western part and the third one (secondary cluster) in the northern part of the KHDSS area. The first cluster had ten villages within a radius of 4.83 km. The number of deaths attributed to other causes in this cluster was seven and the relative risk was 3.20 (p = 0.810). The second cluster was made up of thirty-four

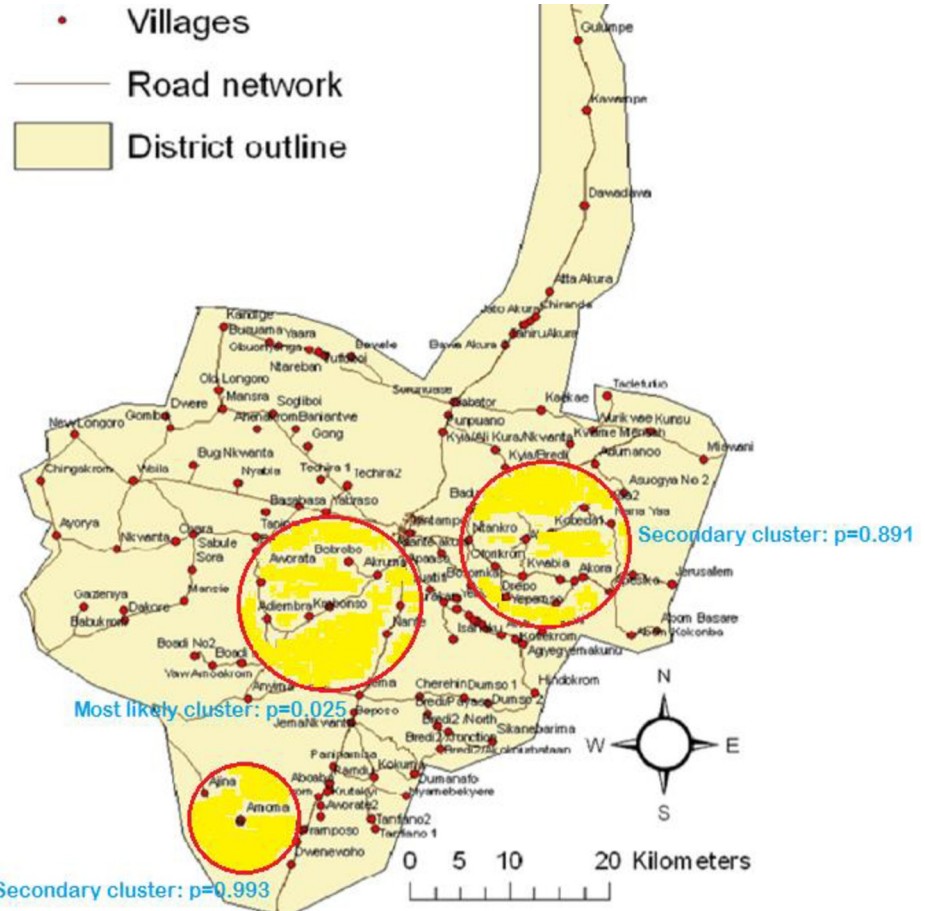

**Fig 4. Most likely and secondary clusters for premature deaths in the Kintampo Districts.**

villages and had a 16.35 km radius. The number of cases identified in this cluster was fifteen and had a relative risk of 2.20 (p = 0.81). There were three villages in the third cluster which had a 3.00 km radius, four cases and a relative risk of 4.35 (p = 0.85). Three potential clusters attributed to other causes of neonatal deaths are depicted in Fig 5.

Two potential clusters were identified for unexplained causes of neonatal deaths; the first cluster (most likely cluster) was in the north-western part and the other cluster (secondary cluster) in the southern part of the KHDSS area. The first cluster had a 3.69 km radius with three villages. The number of cases in this cluster was three with a corresponding relative risk of 15.33 (p = 0.222). The second cluster covered a radius of 0.91 km and consists of two villages. The number of cases in the cluster was three and the relative risk was 7.82 (p = 0.689). The primary and secondary clusters for unexplained causes are shown in Fig 6.

Kintampo Districts' map which forms part of the figures were extracted from elsewhere [24].

## Discussion

The study identified emerging and statistically significant mortality clusters in the Kintampo districts. The most likely cluster for all-cause neonatal mortality was in the south-western part of the Kintampo Districts and contains eleven villages. Of the eleven villages, Nante (32.4%), Ampoma (18.3%), Krabonso (15.5%), Hyireso (15.5%) and Nante Zongo (11.3%) are the

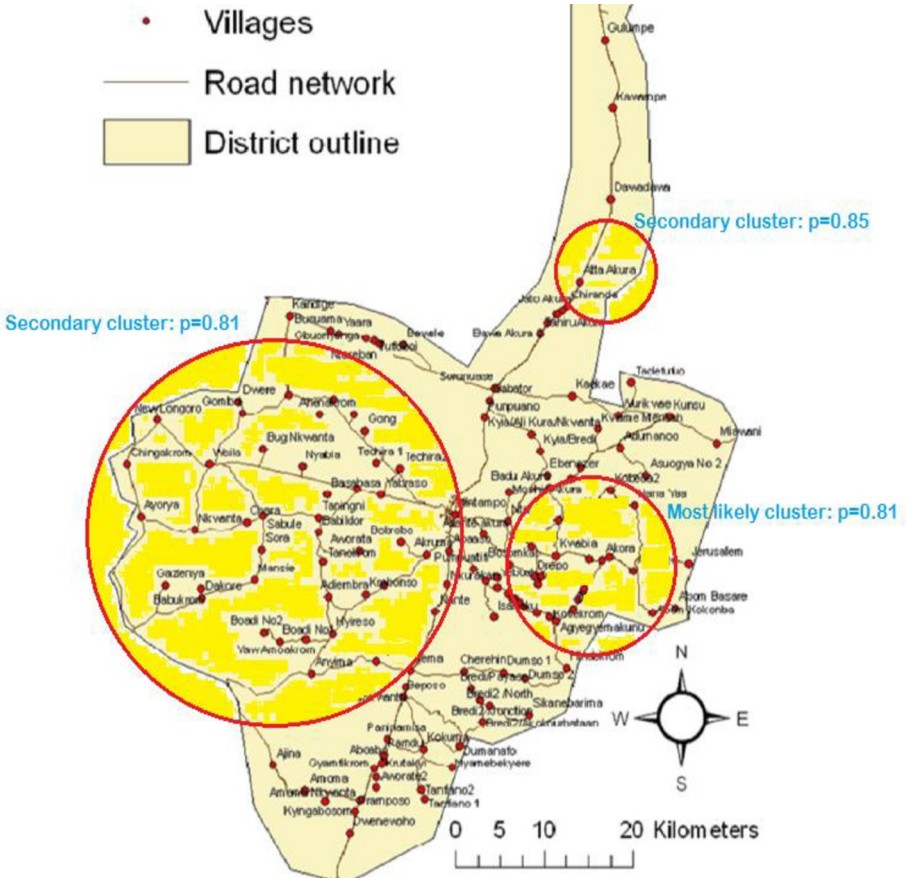

**Fig 5. Most likely and secondary clusters for other causes of neonatal deaths in the Kintampo Districts.**

villages that contributed most to the 71 deaths that occurred in the cluster. Neonatal deaths from Nante, Ampoma and Nante Zongo could be due to lack of well-equipped healthcare facilities such as Neonatal Intensive Care Unit (NICU) and well-trained healthcare personnel such as neonatologists and paediatricians in the district hospital which members of these communities highly patronised. Apart from Nante, Ampoma and Nante Zongo which are close to the district hospital in Jema and are linked to Jema by the main highway, Hyireso and Krabonso are far from Jema and their road linking the main highway to Jema is bad. Means of transport are also not readily available. Perhaps the high proportion of all-cause neonatal deaths in these two villages are due to delay in transporting neonates to the district hospital during emergency situations. The common factor that may be attributed to a higher all-cause neonatal mortality in the five villages is a lower proportion of mothers with secondary or higher level of education. Further analysis of the data revealed that the proportion of mothers with secondary or higher educational level in the five villages constitute only 2.4% of all mothers with that level of education. Neonates whose mothers have secondary or higher level of education are supported by literature to have low risk of mortality [10, 25]. The possible contribution of these category of mothers to neonatal deaths is their inability to adhere to instructions and prescriptions from clinicians, non-adherence to prompt vaccination schedules, and their inability to defy strict negative cultural norms associated with neonatal deaths [11, 26, 27].

With regards to cause-specific neonatal mortality, the most likely cluster with eighteen villages had a statistically significant effect on asphyxiated deaths. Nine of the villages namely

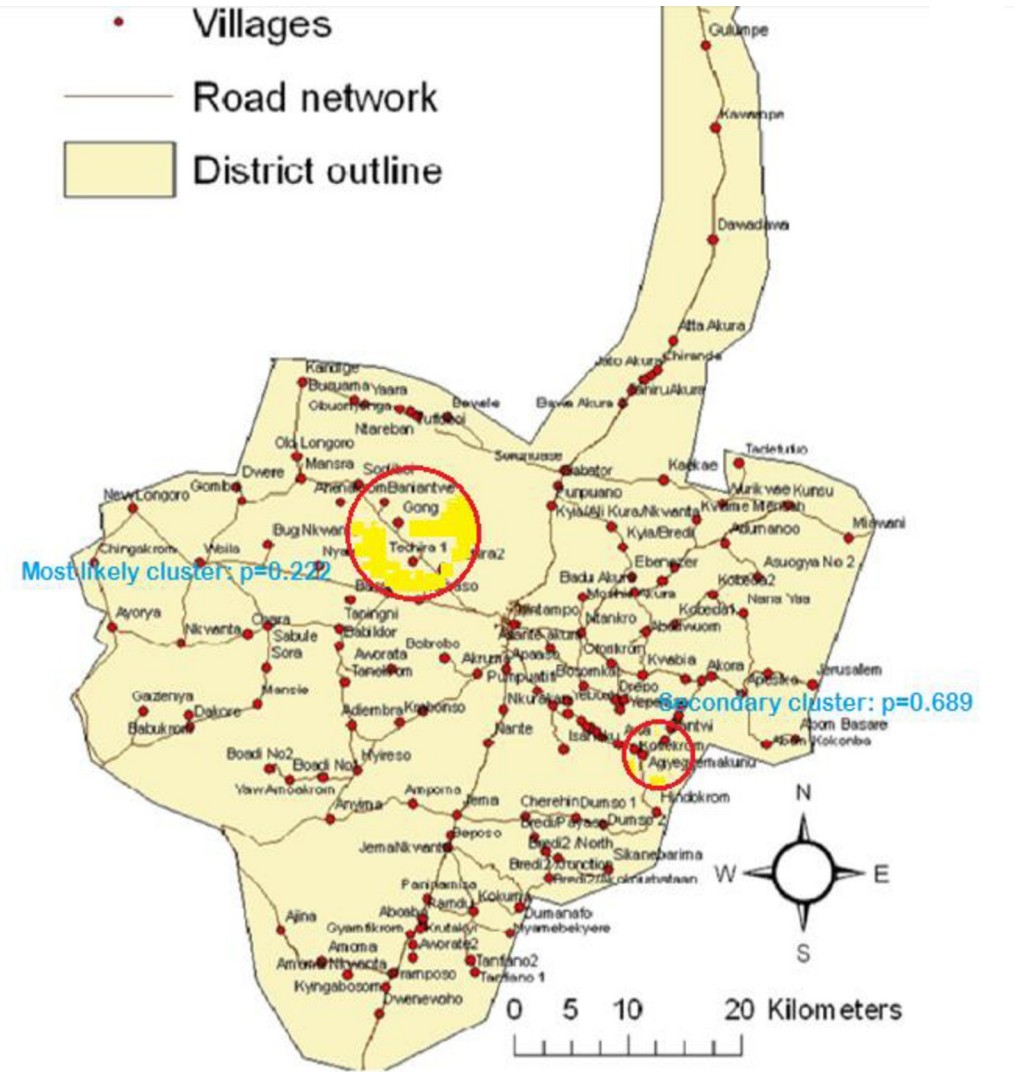

**Fig 6. Most likely cluster and secondary cluster for unexplained deaths in the Kintampo Districts.**

Jema (19.6%), Anyima (16.1%), Pamdu (12.5%), Mansie (8.9%), Ajina (7.1%), Paninamisa (7.1%), Ampoma (7.1%), Krabonso (7.1%) and Hyireso (5.4%) contributed a significant proportion of the asphyxia-related deaths. Ampoma, Krabonso and Hyireso that contributed significantly to all-cause mortality are also included in this cluster. Apart from Jema, Ampoma, Paninamisa and Pamdu which do not have bad road and transportation problems, the others have these problems. Since asphyxia-related deaths usually occur on the day of birth, the possible reason for the villages with unmotorable roads and the lack of transportation might have been delays in reaching the hospital when there were labour complications. Further analysis of the study data supports the afore-mentioned reason since nearly half (48%) of the asphyxiated deaths in these villages occurred in the hospital and this may be due to delay in reaching the hospital. The other probable explanation is the delay in referral of high-risk pregnancies from health centres, CHPS compounds, and traditional birth attendants in some of these villages to the referral hospital in the district. It may also be due to lack of basic neonatal resuscitation

and provision of suboptimal Emergency Obstetric and Neonatal Care (EmoNC) [28] at the referral hospital.

The most likely cluster for deaths attributed to infections has three villages with Nante (50%) and Nante Zongo (40%) contributing most to the ten cases that occurred in the cluster. Neonatal deaths in these villages may be related to inadequate continuum of care and lack of adequate supply of medicines and poor coverage of services that targets neonatal survival [29]. Additionally, health facilities in these villages are not well-equipped and may not be able to effectively treat neonatal infections. Lack of prompt identification of danger signs associated with these infections by mothers may also contribute to these deaths [30].

There was significant clustering for premature neonatal deaths. Villages that contributed high proportions of these deaths are Nante (60%) and Krabonso (30%). Premature babies need intensive neonatal care services for their survival [31] but the villages in this cluster lack such services and will ultimately contribute to these deaths. Other reasons such as lack of well-equipped health facilities, skilled birth attendance, inadequate prenatal care, delay in attending to obstetric complications may be the underlying reasons for the deaths in this cluster. Malaria is a correlate of prematurity [32] and the villages in the cluster are malaria endemic. Therefore malaria could contribute to the underlying causes of the premature deaths.

The most likely cluster for other-related deaths contains villages that are in the outskirts of southern part of Kintampo Districts. Therefore, deaths of this nature may be due to, lack of skilled birth attendants, lack of health personnel, lack of well-equipped health facilities, poor socio-economic status, unmotorable roads, poor housing conditions and lack of transportation.

The most likely cluster for unexplained deaths is in the northern part of the Kintampo Districts. Though not statistically significant, it has the potential for significant mortality clustering. It contains three villages namely Gong, Baniatwe and Techira No. 1 which are in the outskirts of the northern part of the Kintampo Districts. The northern part has few but unequipped health facilities, inadequate health personnel, inadequate skilled birth attendants, and few modern amenities such as electricity and good sanitation facilities. The nearest hospital of these villages in the north is the Kintampo North Municipal Hospital which is far from these villages and moreover transportation is not readily available in these villages during emergency situations. Socio-economic status in these villages within the cluster is also poor and road networks are also poor. All the enumerated factors may lead to clustering for unexplained neonatal deaths.

This study has, however, some few limitations. Using the Kulldorf method has a limitation of including low-risk areas surrounded by high-risk areas in a cluster, since it uses a circular window to scan for high-risk clusters [33]. However, the spatial analysis results for this study identified all the low-risk villages which were surrounded by high-risk villages. Therefore, any future intervention rolled out would be able to identify all the high-risk villages. Neonatal deaths are also likely to be under-enumerated due to the difficulty in the collection of information related to neonatal deaths in developing countries [34, 35]. However, this will be reduced in this study since the study has Community Key Informants (CKIs) who report any events of births and deaths in-between the scheduled household visits. There were some missing records which are also a limitation to this study. Finally, the VA instrument used to determine the causes of death is not as accurate as the autopsy performed in the hospital.

## Conclusions

The study suggests higher risk of asphyxia-related deaths in some cluster of villages in the Kintampo Districts. Also, premature deaths appears to be higher than average in some cluster of

villages in the Kintampo Districts. It will be appropriate to conduct a qualitative study to elicit factors explaining these high neonatal cause-specific mortality clustering.

## Supporting information

**S1 Dataset.**
(DBF)

**S2 Dataset.**
(DBF)

**S3 Dataset.**
(DBF)

**S4 Dataset.**
(DBF)

**S5 Dataset.**
(DBF)

**S6 Dataset.**
(DBF)

**S7 Dataset.**
(DBF)

**S8 Dataset.**
(DBF)

**S9 Dataset.**
(DBF)

**S10 Dataset.**
(DBF)

**S11 Dataset.**
(DBF)

**S12 Dataset.**
(DBF)

**S13 Dataset.**
(DBF)

**S14 Dataset.**
(DBF)

## Acknowledgments

We wish to thank the community members of Kintampo Districts and staff of Kintampo Health Research Centre.

## Author Contributions

**Conceptualization:** George Adjei.

**Formal analysis:** George Adjei, Eugene K. M. Darteh, David Teye Doku.

**Methodology:** George Adjei, Eugene K. M. Darteh, David Teye Doku.

**Supervision:** David Teye Doku.

**Writing – original draft:** George Adjei.

**Writing – review & editing:** George Adjei, Eugene K. M. Darteh, David Teye Doku.

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
