## [Decision Letter · Decision Letter 0]

10 Sep 2020

PONE-D-20-22599

Neonatal Mortality Clustering in the Kintampo Districts

PLOS ONE

Dear Dr. ADJEI,

Thank you for submitting your manuscript to PLOS ONE. After careful consideration, we feel that it has merit but does not fully meet PLOS ONE’s publication criteria as it currently stands. Therefore, we invite you to submit a revised version of the manuscript that addresses the points raised during the review process.

We look forward to receiving your revised manuscript.

Kind regards,

Rohina Joshi

Academic Editor

PLOS ONE

Journal Requirements:

2.We note that the figures  in your submission contain [map/satellite] images which may be copyrighted. All PLOS content is published under the Creative Commons Attribution License (CC BY 4.0), which means that the manuscript, images, and Supporting Information files will be freely available online, and any third party is permitted to access, download, copy, distribute, and use these materials in any way, even commercially, with proper attribution. For these reasons, we cannot publish previously copyrighted maps or satellite images created using proprietary data, such as Google software (Google Maps, Street View, and Earth). For more information, see our copyright guidelines: http://journals.plos.org/plosone/s/licenses-and-copyright.

1.    You may seek permission from the original copyright holder of the figures to publish the content specifically under the CC BY 4.0 license. 

Reviewers' comments:

Reviewer's Responses to Questions

**Comments to the Author**

1. Is the manuscript technically sound, and do the data support the conclusions?

Reviewer #1: Yes

Reviewer #2: Yes

2. Has the statistical analysis been performed appropriately and rigorously? 

Reviewer #1: Yes

Reviewer #2: Yes

3. Have the authors made all data underlying the findings in their manuscript fully available?

Reviewer #1: Yes

Reviewer #2: Yes

4. Is the manuscript presented in an intelligible fashion and written in standard English?

Reviewer #1: Yes

Reviewer #2: Yes

5. Review Comments to the Author

Reviewer #1: This is an interesting paper aimed to identify high-risk clusters for neonatal mortality in Kintampo districts, Ghana. The paper is clearly written and makes an important contribution to the knowledge in the area, describing the methods and application of the identification of clusters of neonatal deaths. l just add a few comments for the authors´consideration.

1. In the title, it would be good to highlight the study takes place in Ghana.

2. The introduction clearly presents the background and objectives of the study. I would suggest to incorporate some estimates of the Global Burden of Disease study in the description of neonatal mortality levels and trends in Ghana and sub-Saharan Africa.

3. The study uses verbal autopsy to ascertain the cause of death. It seems to me the authors used an ad hoc instrument, and the cause of death was ascertained by a panel of physicians, what is OK. Was there any chance to use any of the standardized VA instruments around (e.g. WHO or PHMRC), and analyze the information using any of the automated methods available (Tariff/SmartVA, InSylico, etc.)?

4. The clustering presented by the authors can be very useful in the planning of health services to prevent neonatal deaths. A crucial role in these deaths is the action of the health system, especially in the attention of neonatal emergencies. Is it possible to discuss the results in relation to the quality of care of services available in these clusters, and not only in terms of the risk factors faced by women and children?

Reviewer #2: Thank you for inviting me to review this paper. I think this paper will be beneficial in identifying clusters of neonatal mortality in the Kintampo district of Ghana. This manuscript could be a candidate for publication, but I have some suggestions that I think would improve the quality:

- What are some other potential reasons for particular clusters that have high cause-specific mortality? Distance and maternal mortality seemed to be the only factors mentioned for each cluster. I understand this is a quantitative study, but I found it difficult to understand the difference in cause-specific mortality when the reasons for the differences were essentially the same for each cluster.

- Verbal autopsy should be listed as a potential limitation to these analyses. Predictive performance of VA is not 100% relative to true autopsy and would impact the interpretation of these findings. I also think the fact that verbal autopsy was used as a diagnostic method should be mentioned in the abstract.

- Finally, the figures with the maps are informative. I wonder if it would be possible to have a combined map that identifies the clusters with the highest all-cause and cause-specific mortality. However, I appreciate this may be technically difficult and messy.

6. PLOS authors have the option to publish the peer review history of their article (what does this mean?). If published, this will include your full peer review and any attached files.

Reviewer #1: **Yes: **BERNARDO HERNANDEZ

Reviewer #2: No

---

## [Author Response · Author response to Decision Letter 0]

28 Feb 2021

Dear Sir/Madam,

RESPONSE TO REVIEWERS AND ACADEMIC EDITOR

Many thanks for the invaluable and insightful comments from the Reviewers and the Academic Editor. I think the comments will really help to improve this manuscript. Please find below the point-by-point responses to the useful comments raised by Reviewers and the Academic Editor. Please refer to the file labelled “Manuscript” for the point-by-point responses. The Reviewers’ and Academic Editors’ comments are italised to distinguish them from the Authors’ responses. 

Academic Editor

Please, the URL provided was accessed and the manuscript has now been put in PLoS ONE format.

2.We note that the figures in your submission contain [map/satellite] images which may be copyrighted. All PLOS content is published under the Creative Commons Attribution License (CC BY 4.0), which means that the manuscript, images, and Supporting Information files will be freely available online, and any third party is permitted to access, download, copy, distribute, and use these materials in any way, even commercially, with proper attribution. For these reasons, we cannot publish previously copyrighted maps or satellite images created using proprietary data, such as Google software (Google Maps, Street View, and Earth). For more information, see our copyright guidelines: http://journals.plos.org/plosone/s/licenses-and-copyright.

We recommend that you contact the original copyright holder with the Content Permission Form (http://journals.plos.org/plosone/s/file?id=7c09/content-permission-form.pdf)

Response: Please, the original Fig 1 has been deleted from the manuscript due to editorial issues regarding copyright policy. The manuscript has therefore been revised accordingly and Fig 2 is currently Fig 1, Fig 3 currently Fig 2 and so on. However, a copy of the approved content permission form for the Kintampo Districts’ map which forms part of the current figures has been submitted to this reputable journal.

Reviewer #1

This is an interesting paper aimed to identify high-risk clusters for neonatal mortality in Kintampo districts, Ghana. The paper is clearly written and makes an important contribution to the knowledge in the area, describing the methods and application of the identification of clusters of neonatal deaths. l just add a few comments for the authors´consideration.

Response: Many thanks

1. In the title, it would be good to highlight the study takes place in Ghana.

Response: Well appreciated. The title of the manuscript has been revised to read: “Neonatal mortality clustering in the central districts of Ghana”

2. The introduction clearly presents the background and objectives of the study. I would suggest to incorporate some estimates of the Global Burden of Disease study in the description of neonatal mortality levels and trends in Ghana and sub-Saharan Africa.

Response: Many thanks for the good suggestion. Please, the Authors looked at the Global Burden of Disease paper but realised Hug et al. (2019) global and regional estimates (citation number is 2) were more recent. Therefore, the Authors have expatiated it to include your suggestion (please refer to page 3, lines 61-63 and page 4, lines 68-69).

3. The study uses verbal autopsy to ascertain the cause of death. It seems to me the authors used an ad hoc instrument, and the cause of death was ascertained by a panel of physicians, what is OK. Was there any chance to use any of the standardized VA instruments around (e.g. WHO or PHMRC), and analyze the information using any of the automated methods available (Tariff/SmartVA, InSylico, etc.)? 

Response: Thanks for this comment. Please looking at the data collection period (January 2005-December 2014), the 2016 WHO VA instrument was not available by then. However, currently the Kintampo HDSS is using the afore-mentioned VA instrument and making use of the InterVA software for coding.

4. The clustering presented by the authors can be very useful in the planning of health services to prevent neonatal deaths. A crucial role in these deaths is the action of the health system, especially in the attention of neonatal emergencies. Is it possible to discuss the results in relation to the quality of care of services available in these clusters, and not only in terms of the risk factors faced by women and children?

Response: Well appreciated, the discussion has been revised to include the health system (Please refer to page 17, lines 289-291; page 18, lines 304-310, lines 312-314, lines 316-320; and page 19, lines 324-325, lines 330-331)

Reviewer #2:

Thank you for inviting me to review this paper. I think this paper will be beneficial in identifying clusters of neonatal mortality in the Kintampo district of Ghana. This manuscript could be a candidate for publication, but I have some suggestions that I think would improve the quality:

Response: Many thanks

- What are some other potential reasons for particular clusters that have high cause-specific mortality? Distance and maternal mortality seemed to be the only factors mentioned for each cluster. I understand this is a quantitative study, but I found it difficult to understand the difference in cause-specific mortality when the reasons for the differences were essentially the same for each cluster.

Response: Well appreciated, the discussion has been revised to incorporate the suggestion (Please refer to page 17, lines 289-291; page 18, lines 304-310, lines 312-314, lines 316-320; and page 19, lines 324-325, lines 330-331).

- Verbal autopsy should be listed as a potential limitation to these analyses. Predictive performance of VA is not 100% relative to true autopsy and would impact the interpretation of these findings. I also think the fact that verbal autopsy was used as a diagnostic method should be mentioned in the abstract.

Response: Thanks, the Authors agree with you. This has been stated as a limitation (please refer to page 20, lines 346-347). Verbal autopsy has also been mentioned as a diagnostic method in the abstract (please refer to page 2, line 35).

- Finally, the figures with the maps are informative. I wonder if it would be possible to have a combined map that identifies the clusters with the highest all-cause and cause-specific mortality. However, I appreciate this may be technically difficult and messy.

Response: Your suggestion is a good one and well appreciated but because some of the villages were in the highest all-cause and cause-specific mortalities, it was not feasible when we tried to do it.

---

## [Decision Letter · Decision Letter 1]

9 Jun 2021

Neonatal mortality clustering in the central districts of Ghana

PONE-D-20-22599R1

Dear Dr. ADJEI,

We’re pleased to inform you that your manuscript has been judged scientifically suitable for publication and will be formally accepted for publication once it meets all outstanding technical requirements.

Kind regards,

Kazumichi Fujioka

Academic Editor

PLOS ONE

Additional Editor Comments (optional):

Reviewers' comments:

Reviewer's Responses to Questions

**Comments to the Author**

1. If the authors have adequately addressed your comments raised in a previous round of review and you feel that this manuscript is now acceptable for publication, you may indicate that here to bypass the “Comments to the Author” section, enter your conflict of interest statement in the “Confidential to Editor” section, and submit your "Accept" recommendation.

Reviewer #1: All comments have been addressed

2. Is the manuscript technically sound, and do the data support the conclusions?

Reviewer #1: Yes

3. Has the statistical analysis been performed appropriately and rigorously? 

Reviewer #1: Yes

4. Have the authors made all data underlying the findings in their manuscript fully available?

Reviewer #1: Yes

5. Is the manuscript presented in an intelligible fashion and written in standard English?

Reviewer #1: Yes

6. Review Comments to the Author

Reviewer #1: (No Response)

7. PLOS authors have the option to publish the peer review history of their article (what does this mean?). If published, this will include your full peer review and any attached files.

Reviewer #1: **Yes: **BERNARDO HERNANDEZ

---

## [Editor Report · Acceptance letter]

17 Jun 2021

PONE-D-20-22599R1 

Neonatal mortality clustering in the central districts of Ghana 

Dear Dr. Adjei:

I'm pleased to inform you that your manuscript has been deemed suitable for publication in PLOS ONE. Congratulations! Your manuscript is now with our production department. 

Kind regards, 

on behalf of

Dr. Kazumichi Fujioka 

Academic Editor

PLOS ONE